# Farmers' Market Usage, Fruit and Vegetable Consumption, Meals at Home and Health–Evidence from Washington, DC

**Xiaochu Hu [1],\*, Lorraine W. Clarke [2] and Kamran Zendehdel [3]**

1   Association of American Medical Colleges, Washington, DC 20001, USA
2   Department of Natural Sciences and Engineering, Prince George's Community College,
    Largo, MD 20774, USA; clarkelw@pgcc.edu
3   Center for Sustainable Development, University of the District of Columbia, Washington, DC 20008, USA;
    kamran.zendehdel@udc.edu
\*   Correspondence: xhu@aamc.org

**Abstract:** Using a survey of 440 residents in Washington, DC metropolitan area conducted in 2018, we empirically examined the causal relationship between farmers' market usage and indicators of health, such as fruit and vegetable consumption, meal preparation time, meals away from home, and body mass index (BMI). On average, we found that a one percent increase in farmers' market usage increases consumers' fruit and vegetable consumption by 6.5 percent ($p < 0.01$) and daily time spent on meal preparing by 9.4 percent ($p < 0.05$). These impacts were enhanced by 2SLS models with instrumental variables which indicates causal effects. Farmers' market usage is also associated with decreased amount of meals away from home ($p < 0.05$). We also found qualitative evidence that shopping at farmers' markets improves access to and increases consumption of healthy food. However, we did not find that farmers' market usage has statistical association with grocery shopper's body mass index. Our study established causality that farmers' market usage positively impacts consumers' fruit and vegetable consumption and meals at home. It provided concrete evidence for interventions aiming to increase dietary consumption and promote healthy eating habits through farmers' markets.

**Keywords:** famers market; fruit and vegetable consumption; meals away from home; body mass index; food environment

## 1. Introduction

Farmers' markets, the most prevalent form of direct agriculture marketing, have gained popularity in the United States during the last two decades. Farmers' markets are considered the historical flagship of local food systems [1]. The rise of farmers' markets goes hand in hand with increasing public awareness of healthy eating, and buying local, along with mounting concerns of Americans' unhealthy diet and increasing obesity rate. Farmers' markets are an integral part of the urban rural linkage and have continued to rise in popularity, mostly due to the growing consumer demand for obtaining fresh products directly from farms. Public health advocates and policy makers are also increasingly promoting farmers' markets as a viable source of fresh and healthy produce in urban settings [2,3].

Washington DC has more than quadrupled regional farmers' markets in the last 25 years, the highest per capita growth in the U.S. [4,5]. This increase is in line with the growing national interest in farmers' markets as well. In 2017, there were more than 174 farmers' markets in the Washington, DC metropolitan area, with about one third in the District of Columbia, one third in Northern Virginia, and one third in metropolitan Maryland [6]. In 2020, after COVID-19 started, 42 farmers' markets in the District of Columbia remained open under social safety policies developed by the city's Food Policy Council (Farmers' markets. https://coronavirus.dc.gov/farmersmarkets, accessed on

29 April 2021). Their innate ability to be outside rather than prolonged indoor grocery shopping (recommended by the CDC (Choose safer activities. CDC. https://www.cdc.gov/coronavirus/2019-ncov/daily-life-coping/participate-in-activities.html) (accessed on 20 May 2021)., has allowed for altered, but continuous, use by communities hit hard by the coronavirus pandemic [7]. Their resilience in the face of intense change in the food industry has allowed them to be part of a body of essential businesses in urban areas, as farmers have pivoted to more direct to consumer avenues for their goods [8,9].

Furthermore, farmers' markets are vital for cities' circular economy for food, and an important method to shift the current, linear food system toward a more efficient, circular food system [10]. Farmers' markets, as a public space, have immense potential to demonstrate, promote, and engage businesses and residents with circular economy practices. During the last few decades, farmers' markets have increasingly been supporting local businesses while also attempting to tackle food insecurity and malnutrition in urban areas. However, the challenge of how local circular economy practices, such as waste diversion, sustainable packaging, and energy use, should be incorporated into the rules, regulations, and daily operations of the market itself, is yet to be uncovered. The COVID-19 pandemic has also exposed the vulnerability of global, regional and local food supply chains [11], and has emphasized how farmers and other agricultural industries must adapt in increasingly digital and remote settings. Understanding how essential direct to consumer agriculture like farmers' markets impact human health is important to protecting their consumer base and supporting their business during crises.

Despite a large body of literature on farmers' markets, research evidence on their health impacts remains limited for several reasons. First, information on how farmers' markets affect consumers' health is largely anecdotal and is often based on small scale, temporary pop up market experience reports rather than studies designed with natural control groups [12]. Second, because farmers' markets usage data are hard to come by in any public used data source, many systematically designed studies have had to rely on geographic proximity to farmers' markets as a proxy of usage instead of identifying actual participation or usage [13–15].

Third, few studies have addressed the causality between farmers' market usage and improved health. This is largely due to concerns about endogeneity, as healthy people will seek out healthy food. This could mean they purposely choose to live close to a farmers' market, or farmers' markets launch in affluent and health conscious communities where people are willing and able to afford organic and fresh food. In addition, those who participate in farmers' markets more frequently have been shown to have a shared set of values against industrialized food and pro-environment, which confounds more direct health measurements. Two studies that shed light on this two way relationship are exceptionally insightful: Allcott et al. [16] investigated local fruit and vegetable (FV) consumption change both by observing low income residents in neighborhoods with good food access and by observing how grocery shopping habits change after a new market opens in a food desert area. They found that the improvement of food access explains a mere nine percent improvement in healthy eating, with the rest all attributable to demand and, ultimately, to education. Leung et al. [15] examined the impact of the neighborhood food environment on young girls' BMI using a three year longitudinal design.

To address these gaps in farmers' market research, we conducted a survey that measures consumers' visitation and spending at local farmers' markets and their health behaviors, as well as investigated the relationships between farmers' market usage, dietary and health behaviors, and health outcomes. Our study improves understanding of the relationship between farmers' markets, access to healthy food and health behaviors of participants, and demonstrates the importance of public health policies and programs through farmers' markets which aim to increase FV consumption and promote healthy eating habits.

## 2. Data and Methods

### 2.1. Data Collection

The data used for this research were collected by the GfK Group on behalf of the University of the District of Columbia (UDC). The target population consists of the following: non-institutionalized adults, 18 and older in age, residing in the Washington DC metropolitan area, who are primary grocery shoppers (defined as a person who is responsible for at least half of the household's grocery shopping, including at grocery stores, convenience stores, supermarkets, or other locations). Gfk invited 767 members from its KnowledgePanel, a probability-based web panel designed to be representative of the United States to the survey (the geodemographic benchmarks used to weight the active panel members for computation of size measures include: gender, age, race/Hispanic ethnicity, education, household income, and home ownership status). From 767 surveyors, 522 completed the survey and 440 were validated. A poststratification statistical weight was provided by GfK, using the geodemographic distribution, such as gender, race and ethnicity, education, household income, and state of residency, for the corresponding population obtained from the Current Population Survey and the American Community Survey. These weights were applied in all of the analyses.

Data were collected from 24 January 2018 to 5 February 2018. The questionnaire included questions about individual and household characteristics, usual grocery shopping and dietary consumption habits, and farmers' market usage, including shopping frequency, money spent, reasons for shopping at farmers' market, transportation, etc. Respondents provided answers based on their and their family members' behaviors during the most recent farmers' market season (May to November). The survey was implemented via email.

### 2.2. Variables and Measures

We measured health with four indicators (all self-reported): daily FV consumption, meal preparation time, meals away from home, and BMI. FV consumption is measured by self-estimated "fresh FV consumption per day (cups)". Fruits and vegetables are associated with better health and longer lifespan, and farmers' markets are closely tied to FV consumption, as suggested by past studies [16–20]. Meals away from home is the sum of eat out and carry out meals per week. Meal preparation time and meals away from home have been highlighted as important indicators for people's health, since meals at restaurants (including fast food restaurants and sit down dinning places) are usually high sodium and fat and are associated with negative health outcomes [21]. Similarly, a higher number of meals at home has been shown to reduce daily fat intake by nearly 1%, and sugar intake by 10% daily [22]. We also used BMI as an indicator of participant health outcomes since it is relatively easy to obtain through the survey and commonly used in previous studies on assessing farmers' markets' health impacts [15]. We are aware that BMI has flaws as a prediction of health, and we will discuss that later in the results section.

Our key explanatory variable is farmers' market usage as a percentage of family's total expenditure on food. We asked people their frequency of visiting any farmers' market in the area, and how much money they spend on a typical market day when they do visit. Of all the sampled residents, about 20 percent have never visited any farmers' markets, about 43 percent are frequent customers who visit farmers' markets at least once a month, on average, and about 14 percent are frequent visitors of a nearby farmers' market at least once a week during the region's market season. In terms of spending, 81 percent of the shoppers spend under 30 dollars on a typical market day. Shoppers were also asked about the type of goods they purchase at farmers' markets, as well as their households' average weekly expenditure on food. We assumed the average spending per market day is all on food items for those who reported shopping at farmers' markets. We are aware that this measure is likely an overestimate of purchase of food, since some people purchase non-food items (such as flowers and crafts), but our data do not allow us to separate food purchase from non-food purchase. In addition, we admit that not all food purchases are on fresh produce or healthy food. Although we assume that fruits and vegetables were the

primary purchased items, a recent study indicates that even purchasing and consuming locally produced cheese, meats, and pastries was beneficial to health due to the reduction in processing [23].

Using the farmers' market visitation frequency, average spending, and the percent of food expenditure, we generated total annual spending on food from farmers' markets. We then used the percent of farmers' market purchases out of total food spending as our measure of farmers' market usage. Considering most of this area's farmers' markets are seasonal, we equate visiting frequency of "more than once per week" as 40 times per year, "once a week" as 30 times, "once every 2–3 weeks" as 15 times, "once every month" as 7 times, "once every 2–3 months" as 3 times, "less than once every 2–3 months" as 1 time per year and "never" as 0 times. Descriptive statistics of all variables used in the regression analysis are shown in Table 1.

**Table 1.** Key statistics for variables included in analysis (farmers' markets users only, weighted).

| Group | Variable | Mean | St. Dev | Min | Max | N |
|---|---|---|---|---|---|---|
| | Weekly meals away from home | 3.6 | 3.2 | 0 | 22 | 360 |
| Health | BMI | 28.0 | 6.0 | 17.0 | 65.0 | 354 |
| Indicators | Daily time spent in preparing meal (mins) | 44.3 | 27.5 | 0 | 200 | 362 |
| | Daily fresh FV consumption (cups) | 3.4 | 2.1 | 0 | 18 | 360 |
| Farmers' market Usage | FM usage (percent of FM purchase in total household food purchase) | 4.6 | 7.36 | 0.04 | 48.07 | 358 |
| | Male | 0.42 | – | – | – | 364 |
| | Age | 46.4 | 20 | 89 | | 364 |
| | Bachelor's degree or higher | 0.60 | – | – | – | 364 |
| Control | Non-Hispanic White | 0.52 | – | – | – | 364 |
| Variables | Non-Hispanic Black | 0.20 | – | – | – | 364 |
| | Non-Hispanic Asian | 0.12 | – | – | – | 364 |
| | Hispanic | 0.14 | – | – | – | 364 |
| | Living with children under 18 | 0.24 | – | – | – | 364 |

Source: UDC Farmers' market Usage survey, 2018. Note: Variables that are not normally distributed are transformed to logged forms in the regression analysis.

### 2.3. Empirical Strategy

First, to quantify the relationship between farmers' market usage and shopper's health behaviors, we used four OLS regressions to examine the relationship between farmers' market usage and the health indicators. This relationship was calculated using the following equation:

$$\text{FV} = + \text{FMusage} + \beta_2 X + \text{Error} \qquad (1)$$

where FV is logged cups of FV each person consumes per day, regressed on *FMusage* (farmers' market usage) captured by the logged percentage of total annual spending at farmers' market in a household's annual food spending. X represents the respondent's demographic and socioeconomic characteristics, including age, gender, race, household income, and number of children under 6. Similarly, weekly meals away from home, weekly meal preparation time, as well as BMI were regressed in this manner.

We then applied 2SLS regression with instrumental variables (IVs) to address potential endogeneity. Endogeneity was a concern here because, in general, people who eat large amounts of fruits and vegetables and spend much time cooking at home may also shop more frequently at farmers' markets. Considering these concerns, we used additional information available in our dataset to construct potential IVs. These include respondents' zip code, method of transportation to farmers' markets, number of farmers' markets they usually visit in Washington, DC metropolitan area, and reasons for shopping at farmers' markets.

Our first IV candidate was the distance between the centroid of shoppers' zip codes to the closest farmers' market. Using distance as an accessibility proxy is common in this type of evaluation. The assumption is that a shorter distance would encourage people to go to the market, but not directly affect health behaviors or outcomes. However, we found no correlation between this IV candidate (distance) and our key independent variable, farmers' market usage (i.e., it did not pass a first stage estimate of the two stage least squares (2SLS) analysis). One obvious explanation is that calculating the distance to farmers' markets more accurately is not possible without residential addresses, a variable not described in our data. The distance from the zip code centroid to the closest market is not granular enough, as in this area many zip codes cover large regions and the density of farmers' markets is high.

Our second IV candidate was based on the method of transportation to farmers' markets: About 25 percent of customers reported that they usually walk to a farmers' market, as opposed to driving or taking public transportation. Therefore, we used "walking to market" as a market accessibility indicator, assuming people who walk to markets visit and use farmers' market more frequently. However, we did not find a significant correlation between walking and market usage either.

In addition, for either of these two geography based IV candidates to work, we would need to assume that geographic proximity to farmers' market is independent of people's health. However, in reality, farmers' markets can choose to open where they see a client base, and people who support the value of farmers' markets may choose to live close to one [24,25].

Our third IV candidate derived from the question "How many markets do you usually visit in your area?". Of all the farmers' market shoppers, three in ten indicated that they visit two markets in their area. We then created the binary variable of "whether someone visits two or more markets" as a proxy of market accessibility and included it as an IV in the 2SLS regression.

Our last IV candidate was an indicator for a "non-health incentivized" customer. All respondents were asked to choose three top reasons for visiting farmers' markets (see Figure 1 for a frequency of all the reasons). We created a binary variable separating customers who shop at farmers' market for socioeconomic reasons (including "to contribute to the local economy", "location/convenience", "helping the environment", "for the community atmosphere", "better price", "accepts my SNAP/WIC/other food assistant benefit") and those who shop for nutrition and health reasons (including "higher nutrition value", "taste", "larger selection", "freshness and appearance"). As most respondents choose multiple reasons for their visit and often overlap between these two broader categories (nutrition/health incentivized and socioeconomic incentivized), we calculated a ratio of socioeconomic reasons over health reasons, then arbitrarily set a cut off for a dummy variable indicating "non-health incentivized" customers. This dummy variable was used as an IV in the 2SLS regression.

All statistical analysis was conducted using Stata 14.

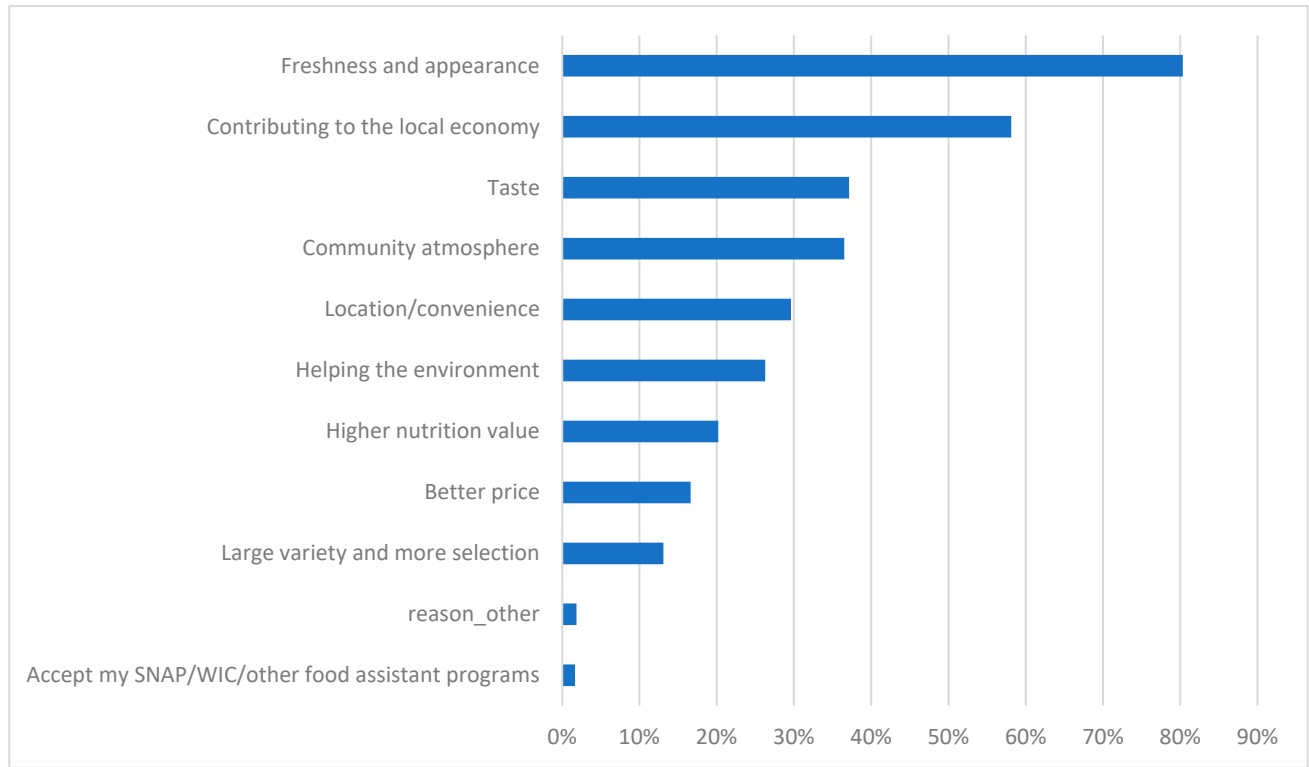

**Figure 1.** Reasons for visiting farmers' markets (excluding those who never visited) n = 365.

### 3. Results

#### 3.1. Link between Shopping at Farmers' Market and Shopper's Health Behaviors

In the survey, respondents were asked to rank their agreement (1—strongly disagree to 5—strongly agree) to the following three statements on a 1–5 Likert scale to evaluate whether a farmers' market has affected their access and consumption of fruit and vegetable. Note that we phrased the statements emphasizing the causality of farmers' market to increased access and increased consumption of the shoppers and their family members.

- Statement 1: "It is easier for me to purchase fresh produce during the market season **because of** the farmers' markets" (increased access)
- Statement 2: "I consume more fresh produce during the market season **because of** the farmers' markets" (increased consumption of shoppers)
- Statement 3: "My family members consume more fresh produce during the market season **because of** the farmers' markets" (increased consumption of family members).

Figure 2 illustrates the frequency of the above qualitative evidence by farmers' market's visitation frequency. Respondents' degree of agreement with these statements increases with farmers' market visitation frequency. More frequent farmers' market shoppers have reported that the market is most helpful in terms of increasing access and consumption of fresh produce for both the primary shopper and their family members. For example, for weekly farmers' market shoppers, 46 percent strongly agree, and another 31 percent agree to the statement that "It is easier for me to purchase fresh produce during the market season **because of** the farmers' markets". This offers intuitive evidence that farmers' markets have a positive impact on fresh produce access and consumption by the grocery shoppers as well as their family members.

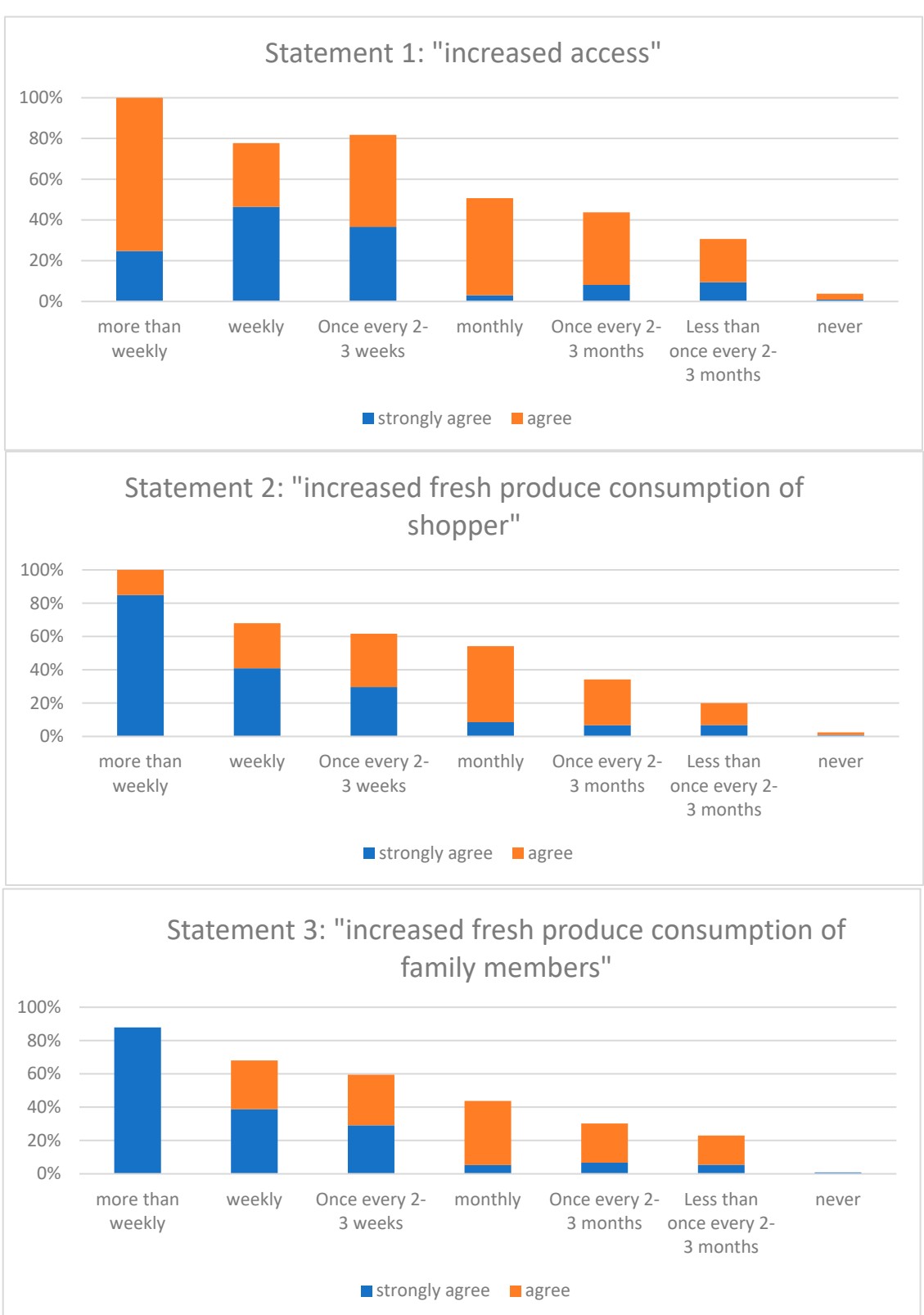

**Figure 2.** Percent of "Strongly agree" and "Agree" to each statement, by visitation frequency. (N = 440).

### 3.2. Statistical Analyses Results

Columns 1–4 in Table 2 present the regression results using Equation (1) without IV. For each regression, the best model results are reported. Weights are controlled in

all regressions. Column 1 shows that farmers' market usage, as a percentage of total food purchase, is significantly associated with an increase in total consumption of FV. With everything else controlled, for each percent increase in farmers' market usage, FV consumption increases 6.5 percent. In addition, more educated people and females tend to eat more FV. Education is positively associated with FV consumption. Compared to non-Hispanic whites, non-Hispanic blacks consume less FV.

**Table 2.** Farmers' market usage and health outcomes regression results (std. err linearized).

| | (1) | (2) | (3) | (4) | (5) | (6) |
|---|---|---|---|---|---|---|
| | FV Consumption (log) | Meal Preparation Time (log) | Meals Bought | BMI (log) | FV Consumption (log)-with Market Accessibility as IV | Meal Preparation Time (log)-with Market Accessibility as IV |
| Farmers' market usage (log) | 0.065 *** | 0.094 ** | −0.271 ** | 0.008 | 0.254 ** | 0.377 * |
| | (0.021) | (0.042) | (0.135) | (0.008) | (0.127) | (0.212) |
| Bachelor's degree or higher | 0.195 ** | 0.093 | 0.401 | −0.068 ** | | |
| | (0.077) | (0.181) | (0.440) | (0.033) | | |
| Education level | | | | | 0.038 | −0.052 |
| | | | | | (0.040) | (0.066) |
| Age | −0.002 | −0.001 | −0.015 | 0.002 ** | −0.004 | −0.007 |
| | (0.002) | (0.004) | (0.014) | (0.001) | (0.003) | (0.006) |
| Male | −0.157 ** | −0.126 | 0.045 | 0.070 ** | −0.085 | 0.016 |
| | (0.079) | (0.174) | (0.498) | (0.031) | (0.095) | (0.196) |
| Non-Hispanic Black (Non-Hispanic White as default) | −0.222 ** | −0.256 | 1.136 | 0.072 ** | | |
| | (0.112) | (0.230) | (0.689) | (0.032) | | |
| Non-Hispanic, Other (Non-Hispanic White as default) | −0.356 *** | −0.010 | 0.808 | −0.011 | | |
| | (0.135) | (0.235) | (0.801) | (0.054) | | |
| Hispanic (Non-Hispanic White as default) | −0.034 | −0.175 | 0.558 | 0.036 | | |
| | (0.152) | (0.342) | (0.842) | (0.061) | | |
| Non-Hispanic, 2+ Race (Non-Hispanic White as default) | −0.510 * | 0.080 | 2.716 * | 0.041 | | |
| | (0.308) | (0.197) | (1.626) | (0.106) | | |
| Household income | 0.006 | 0.012 | −0.084 * | 0.002 | 0.017 | 0.039 * |
| | (0.009) | (0.015) | (0.050) | (0.004) | (0.014) | (0.023) |
| With children | 0.050 | −0.296 | 0.476 | 0.038 | 0.206 | −0.054 |
| | (0.101) | (0.252) | (0.690) | (0.040) | (0.150) | (0.312) |
| constant. | 0.810 *** | 3.309 *** | 4.988 *** | 3.300 *** | 1.690 ** | 5.552 *** |
| | (0.239) | (0.507) | (1.563) | (0.110) | (0.853) | (1.273) |
| N | 354 | 355 | 355 | 348 | 353 | 354 |
| R-Squared | 15.46% | 10.17% | 8.77% | 11.89% | 18.87% | 19.02% |

Source: UDC Farmers' market Usage survey 2018. * $p < 0.10$; ** $p < 0.05$; *** $p < 0.01$.

Column 2 in Table 2 shows that farmers' markets usage is significantly associated with an increase in daily meal preparation time. All else being equal, for each percent increase in farmers' markets usage, daily meal preparation time increases 9.4 percent. The other control variables do not seem to have much influence on meal preparation time. Column 3 shows that farmers' markets usage is significantly associated with a decrease in meals away from home (including carry out and eat out), as we would expect given the fixed sum of meals that people eat.

Column 4 in Table 2 shows that there is no statistically significant relationship between farmers' markets usage and BMI. Being younger, female and more highly educated translates into lower BMI while African Americans tend to have higher BMI. We also tested the predicting ability of BMI using three of our other dependent variables. FV consumption, meal preparation time and meals away from home are interdependent and they all could affect BMI. Meal preparation time is negatively associated with BMI (those that spend longer preparing food have healthier BMIs), though FV consumption and meals away from home do not have a significant relationship with BMI.

Column 5 and 6 in Table 2 use the same variables as Column 1 and 2 (race as a factor variable is not used in the 2SLS regression, and education is used as it results in larger F-statistics in first stage estimates), but here the key explanatory variable (farmers' market usage) is instrumented by market accessibility (whether someone usually visits two or

more markets) using 2SLS regression. Table 3 shows the first stage estimates of this 2SLS. The F-statistic of the first stage estimates for FV consumption and meal preparation time are larger than 12, sufficiently passing the threshold of instrumental variable qualification (10.0). In all three regressions, market accessibility is significantly and positively correlated with farmers' market usage, as expected. Results from Column 5 and 6 show that 2SLS analysis confirmed that farmers' market usage is significantly and positively correlated with both FV consumption and daily time spent in meal preparation, and the magnitude of these effects is also enhanced compared to those in the OLS regression: a 1 percent increase in farmers' market usage is associated with 25 and 38 percent increases in FV consumption and daily meal preparation time, respectively.

**Table 3.** First state estimates using two different IVs—market accessibility and socioeconomically incentivized customers.

| Independent Variable Instrumented: Log Farmers' Market Usage | Dependent Variable | | | |
|---|---|---|---|---|
| | Statistics | (log) FV | (log) meal preparation time | (log) meals bought |
| Instrument: whether someone visits two or more markets | Coefficient | 0.75 *** | 0.74 *** | 0.64 ** |
| | F-statistics for IV | 12.16 | 12.15 | 8.84 |
| | N | 353 | 354 | 357 |
| Instrument: non-health incentivized | Coefficient | −0.119 | −0.15 | −0.252 |
| | F-statistics for IV | 9.49 | 10.04 | 6.11 |
| | N | 293 | 294 | 297 |

$** p < 0.05$; $*** p < 0.01$.

Lastly, we used "non-health incentivized" customers as an IV. The first stage analysis results are also shown in Table 3. Only the model using meal preparation time as a dependent variable narrowly passed the first stage estimates: F-statistics exceed 10.0, and socioeconomically incentivized customers are less likely to visit the farmers' market (but insignificant). In the second stage analysis, farmers' market usage is not significantly correlated with meal preparation time. As noted above, since many respondents chose both health related and socioeconomic reasons for visiting farmers' markets, the cutoff is arbitrary. Using a more divided cut would possibly increase the instrumental variable strength, but at the same time we are faced with a smaller sample size. Although this instrumental variable candidate did not work well, it points to a new direction for addressing the endogeneity concern for future studies and survey design.

The magnitude of the impact of farmers' market usage on FV consumption and daily meal preparation time, especially in the 2SLS models, is large. We also recognize that our measure of farmers' market usage is extrapolated through many steps of logical assumptions from the direct information respondents gave—"frequency" and "average spending per visit"—and both in intervals. Its accuracy also depends on the self-reported total household spending on food, since we used a percent of farmers' market spending out of total household spending on food. Our healthy dietary measure, FV consumption, is measured in the absolute quantity, and does not take into account their proportion in one's diet. Meals away from home is not a perfect measure of healthy eating either, since people who are perfectly health conscious can simply take out or eat out healthy food. We, therefore, regard the magnitudes as suggestive, rather than definitive.

## 4. Discussion and Conclusions

Our research confirms that frequent shopping at farmers' markets is significantly associated with an increase in consumers' FV consumption, daily time spent on meal preparation, and decrease in the amount of meals away from home. Different IV approaches were applied and enhanced these relationships.

In one major finding, we did not find a significant association between shopper's BMI and farmers' market usage. Some studies have found a relationship between healthy food access, like farmers' markets, and weight related measures like BMI, but we note these

studies were either cross sectional [13,14,26,27], leading to the endogeneity issue where healthy people may choose to live in neighborhoods with farmers' markets, or relied on geographic proximity or access as a proxy of farmers' market usage or participation [15]. Indeed, a major issue with healthy food proximity studies in relation to weight related health measures is the lack of attention to the residential food decisions and socioeconomics of neighborhoods [25,26]. Data shows that BMI is of little consequence if participation in markets was measured or more direct measurements of neighborhood food environment were measured to address endogeneity [15,19,28]. The lack of linkage is possibly because BMI has flaws in predicting people's health due to its simplicity, as it does not take frame size or muscle mass into account [29]. More direct measures of health, such as blood pressure, body fat percentage, and mental well-being, were outside the scope of this study but can be directly linked to increased fruit and vegetable consumption [30–33]. FV consumption itself has been linked to farmers' markets and health via sponsored educational programs [34], improved neighborhood participation in markets [35,36], and farmers' market "prescription" vouchers for vegetables [37]. These findings indicate that oversimplified health measures like BMI may be advised against in regional studies, as there is no clear progression or control for muscle vs. fat weight in measuring health. Instead, actions that support healthy lifestyles (like FV consumption and meals at home) are better at estimating health.

Increasing fruit and vegetable consumption itself, outside of BMI, has been shown to have different and more direct positive impacts on health. These benefits include a reduction in cancer occurrence and mortality [34–36], reduced heart disease occurrence [33], and significantly reduced blood pressure in middle aged adults [33,38]. Produce and other market foods sold via local agriculture have been shown to be more nutrient rich than those in normal markets [39]. In the DC metropolitan area, the highest risk individuals for cancer, heart disease, and high blood pressure are black and low-income [40], and are also those that our study identified as having the lowest FV consumption and participation in or access to farmers' markets. The causative reasons for this disparity are likely complex, as fewer farmers' markets exist in low income wards and neighborhoods of the DC region and low-income individuals may have reduced access to markets due to lack of transportation or racial discrimination in those neighborhoods [41]. Adding to this complexity is how education and socioeconomics influences participation in and demand for farmers' markets [16]. Even so, our data insinuate that greater access to and participation in farmers' markets (particularly in low income neighborhoods) could improve a broad variety of health outcomes, especially for at risk groups who consume very little FV, an outcome strongly supported by the literature [12,18,20,22,34].

Our finding of increased time spent in preparation and meals eaten at home as a result of shopping at farmers' markets is unsurprising, as most farmers' market food products are not ready to eat. Even for lower income households, this pattern persists, and the USDA's Thrifty Food Plan (a healthy diet at low-cost) relies heavily on home cooking and not eating out [42]. However, it is also a complex socioeconomic puzzle. Those who eat out more and prepare less in their kitchen may be those who work long hours at a low paying job or have reduced cooking skills [43]. Teasing out how much of the farmers' market participation in certain groups is linked to access versus time investment is a difficult question not answered through our simple survey. Indeed, free time to invest in cooking may be more explanatory than just access to farmers' markets in a person's neighborhood [44]. Additionally complex, the amount of spent on meal preparation at home and decrease in carry out and dining out may benefit the whole family's health instead of just the direct grocery shopper. Although our statistical analysis did not focus this, our survey results offered qualitative evidence that show farmers' markets visits have a positive impact on family members FV consumption.

Our research has limitations that call for future data collection and further investigations: first, as the qualitative evidence suggests, we were only able to statistically examine the association between farmers' market usage and the family members' FV consumption,

meal preparation time and BMI. Further research is needed to directly measure the association between shopping at farmers' markets and family, particularly children's, health. We did not collect information on how food purchased at the farmers' market was distributed among family members and only collected information on health outcomes from the primary grocery shopper. Second, our measures of health and health behaviors—BMI, FV consumption, meal preparation time and meals away from home—are items that could be easily measured via a survey, rather than a comprehensive map of possible paths through which farmers' markets can impact shoppers' health. Other paths, investigated by various studies, include the distribution of special nutrition supplement programs [45] and an increase in customer's physical activities [46]. Third, although our farmers' market usage is already an improvement from geographic approximation, our measure of usage is still crude. In the survey region, most farmers' markets run from May to November. As our survey was conducted in the off-season, our data could suffer from potential bias, as we are relying on people's memories or planned participation. In addition, due to limitations in question detail, our data do not differentiate between food purchase from non-food, or healthy food from unhealthy ones. The key explanatory variable, farmers' market usage as a percentage of family's total expenditure on food, further depends on the accuracy of self-estimated weekly spending on food. More detailed data on farmers' market expenditure would be more accurate in this regard. As more transactions at farmers' markets become digitized (a progress accelerated by COVID-19), analysis of itemized grocery receipts at farmers' markets may be possible soon.

The key strength of our study was the carefully designed consumer survey, which has direct and complex measures of farmers' market usage. This is an important improvement from previous studies on this topic, which relied on the geographic proximity to farmers' markets as a proxy of usage. Second, our data include additional information about farmers' markets usage, such as transportation and shopping reasons, and this allowed us to explore innovative instrumental variable ideas that help address the complex endogeneity issue. Finally, we sought evidence beyond traditional health measures like BMI, and included healthy dietary and eating habit measures. Once these relationships are established, these factors are less compound and more actionable for nutrition and health policymakers to introduce new preventative public health programs and interventions. Allowing a direct comparison with previous studies by including analysis using BMI, we also point to the need for more direct measurements of health and away from "simple" measures of weight ratios. As noted above, increasing FV consumption and meals away from home can improve overall regional health via farmers' markets or other direct to consumer agricultural programs. Coupling programs like "prescriptions" for vegetables at farmer's markets, doubling the value of SNAP at local markets for FV purchases, and introducing educational nutrition and cooking programs with increased outreach to low-income communities could greatly expand the impact of farmers' markets on health outcomes in these impoverished communities [12,14,31,34,45]. In particular, our research emphasizes the need for increased direct market grocery access for low FV consumption populations. Considering the uneven distribution of farmers' markets in the area, improving outreach to underrepresented groups and coupling farmers' markets with other interventions (like food budgeting and nutritional education) could improve health outcomes beyond what was found in this study. Small group longitudinal studies of direct health outcomes (including a more direct accounting of BMI change over time, blood pressure, and FV consumption) would also be a good future direction to confirm and expand on our findings.

**Author Contributions:** Conceptualization, X.H. and L.W.C.; Data curation, X.H.; Formal analysis, X.H.; Funding acquisition, X.H., L.W.C. and K.Z.; Investigation, X.H.; Methodology, X.H.; Project administration, X.H. and K.Z.; Resources, X.H.; Software, X.H.; Supervision, X.H.; Validation, X.H.; Visualization, X.H.; Writing—original draft, X.H. and L.W.C.; Writing—review & editing, X.H., L.W.C. and K.Z. All authors have read and agreed to the published version of the manuscript.

**Funding:** This research was made possible by The National Institute of Food and Agriculture's Seed Grant "Farmers Market's Impacts on Food Security, Regional Economy and Diet" (DC-0042016).

**Institutional Review Board Statement:** The study was conducted according to the guidelines of the Declaration of Helsinki, and approved by the Institutional Review Board of The University of the District of Columbia. Approved in 2017.

**Informed Consent Statement:** Informed consent was obtained from all subjects before participating in the survey.

**Acknowledgments:** We would like to thank Steve Payson at the Bureau of Economic Analysis for reviewing the survey instrument and manuscript, and Bruna Pinto for her excellent assistance in spatial analysis. We also thank the participants for their suggestions and comments at Society of Government Economists conference in DC, 2018 and at Southern Economics Association annual meeting in DC, 2018. The findings and conclusions in this paper are those of the authors and do not necessarily represent the views of NIFA USDA or our organizations.

**Conflicts of Interest:** The authors declare no conflict of interest.

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
