# Peer review of "Farmers’ Market Usage, Fruit and Vegetable Consumption, Meals at Home and Health–Evidence from Washington, DC"

_sustainability, doi:10.3390/su13137437_

Round 1
Reviewer 1 Report
Authors honestly recognize the limits of the analysis. They are methodological as well as conceptual constraints of the knowledge improvement in this field. Positive outcomes are the proposal of some behavioral indicators of consumers, such as the food expenses weight made in farmers' markets or farmers' markets accessibility. The most important result, in my opinion, is the demonstration of the lack of relationship between farmers' market frequentation and BMI. Finally, we have found someone who states it in such a direct and unequivocal way! BMI is a weak explanatory index of food consumption behavior and the link between farmers' market usage and healthy food consumption models is still to be demonstrated. To be honest, the other results produced by the paper are purely trivial and dangerous. Yes, because it is a true danger to spread a message assigning inappropriate roles to farmers' markets. They are a link between the city and the country, where relational goods are produced and consumed and urban people find fresh as well as convenience food when farmers find a good economic solution to their market problems. Please, load on farmers' markets neither health nor environmental functions. Farmers' markets are not the answer, they are not more efficient than the global food system, and the sentence written in the introduction "post-COVID era calls for a shorter, more locally focused food supply chain" is not true. We all are grateful to the global food chain for having fed us during the pandemic period. Farmers' markets closed (look at the data: several farmers' markets did, conventional food resellers did not). They are not more efficient: the food miles are repeatedly downgraded from absolute truth to fantasy legend (see Saunders et al., 2010 or Weber, 2008 for example). Although I do not agree with the introduction and some conclusion statements, the scientific approach of the paper is correct preferably if some caveat is added.
Author Response
Response:
1. We thank the reviewer for recognizing our finding of the non-significant association between farmers market usage and BMI. BMI has too often been held up as the end all be all for health and there are far too many outside factors to assign solely to participation in farmers’ markets. As noted by reviewer 2 and bolstered by this note, we have added new emphasis to the BMI information in the discussion (beginning page 10, line 29). In particular, we note its limitation in a regional study without direct health measurements and issues with endogeneity in studies that have BMI linked to food access.
2. As to the other comments, we strongly believe that our results show a clear and important association between farmers market usage and healthy behaviors, particularly of increased fruit and vegetable consumption, meals cooked at home and time spent preparing meals. Our use of these specific metrics is thoroughly defended in the introduction (page 4), as they have been used by a myriad of researchers to measure and support health outcomes. As noted on page 10-11, starting at line 39, we note that farmers markets specifically support health indices like reduced blood pressure and increased mental well-being (Bharucha et al. 2020, Trapl et al. 2020) through the mechanism of increased fruit and vegetable consumption. Investment in cooking time and increasing meals at home is linked to healthy eating (USDA, Park et al. 2011), and can be specifically linked to purchasing raw ingredients at farmers markets (as noted in page 4, line 8-10 and page 11, line 19-31).
Farmers’ markets are clearly related to health behaviors and do shorten the food chain, important environmental functions. We point the reviewer to their use of food miles to measure environmental benefit, while our paper does not mention this, merely the directness of agricultural purchase. Our outcomes point to new directions for future research. In particular, this research can now be expanded locally to do focus groups and identify causal factors in participation, clearly linking to the overall trends. We can then directly link those factors with FM usage. We have expanded our discussion to include clearer future directions in research and ways to bolster the health benefits of farmers markets.
4. While we believe that farmers’ markets play a strong health role, we agree with this reviewer’s assessment of our “post covid era” comment, as it does overstate how farmers’ markets and other direct to consumer agricultural endeavors fared during the pandemic. It is true that many markets closed and were extremely limited due to the pandemic, particularly those that could not adapt to the remote status or were limited in online factors. Still, it is true that direct to consumer agriculture is in high demand, even during the pandemic (Schmidt et al. 2020). We have removed that sentence and added in clarity about how farmer’s markets (especially in the study area) have fared during the pandemic and indicated that understanding their health role is important to supporting farmers’ markets in this new post covid era.
Changes made:
- We emphasized this non-significant finding in the conclusion and discussion part and cited other studies that contradict this result. (p 10 line 29-47, p 11 line 1- 11)
- We have deleted statement "post-COVID era calls for a shorter, more locally focused food supply chain" and rewritten this part. (p 2 line 32-35). Note the addition of the Schmidt et al. citation that indicates that direct to consumer increased during the pandemic.
- We have added in studies linking increased participation in farmers markets and other local food markets and health outcomes, focusing on FV consumption (pg 10, line 36-47)
- Future outcomes and policy and outreach recommendations for improving health outcomes of farmers markets are outlined more clearly at the end of the paper (page 12, line 6-22)
Reviewer 2 Report
One obvious missing reference:
(2016) Economic and Environmental Drivers of Fruit and Vegetable Intake Among Socioeconomically Diverse Adults in Vermont, Journal of Hunger & Environmental Nutrition, 11:2, 263-271, DOI: 10.1080/19320248.2015.1128862
The "circular economy" is underdeveloped but not essential. Elaborate or remove
You seem to assume that all food at farmers market of FV: what about meat, eggs, bread? Is there a way to tease this out or qualify it?
I'd like to see more detail in the IVs used
With the flaws you cite, does the BMI analysis even make sense to include?
"Data" is a plural noun
Can you posit implications for outreach, policy etc.?

Author Response
Response:
We believe the BMI analysis is also very important. It has been used as a ultimate measure of health in many public health studies and we believe finding no association between FM usage and BMI is an important finding, contrasting to what many other studies have concluded.
We also would like to mention that this part’s importance is required to be brought out more by another reviewer. But we hope this clarification will satisfy both reviewers.
Changes made:
- We have elaborated the statement on circular economy. (Page 2 Line 25-32)
- We are not able to quantify percent of food purchase from non-food purchase. We are aware that not all food purchase are healthy food. This is mentioned in variable description (p4 line 23-27) and as a limitation of the study (p11 line 47-50). We do cite two new studies that show that minimally processed non-FV food from local markets is markedly healthier, and produce/prepared foods from local markets was more nutrition dense (pg 4, line 31-33, and p11, lines 6-7)
- Added Conner & Bernice R. Garnett. (citation #19, p 4 line 3)
- We have added implications for outreach and actional policy recommendations in the discussion part (page 12 line 8-14)
- "Data" verb changed to a plural.
- Unfortunately, our data do not allow us to separate food purchase from non-food, or fresh produce from meat, egg, etc. and this has been one of the limitations in the discussion part. Although we assume that fruits and vegetables were the primary purchased items, a recent study indicates that even purchasing and consuming locally produced cheese, meats, and pastries was beneficial to health due to the reduction in processing (Migliaretti et al 2020) This is discussed in page 4 line 26-33.
- IV construction part has been clarified and revised with more details. (page 5 line 23-47, page 6 line 4-10, and page 7 line 1-7)
Reviewer 3 Report
I read the work with interest. It seems like a good topic to me. The paper, following a correct approach, is quite clear and easy to read.
Going into the merits of the paper, I note that:
- The title does not perfectly reflect the content; it would be appropriate to insert "in the USA";
- The research question is not clear enough;
- The data analysis is for me quite complete and correct;
- Discussion and Conclusion must be better connected with the reference literature;
The literature is full-bodied and quite complete. However, the authors could improve the work by reading the following recent references:
Cicia, G., Furno, M., Del Giudice, T., Do consumers’ values and attitudes affect food retailer choice? Evidence from a national survey on farmers’ market in Germany, 2021, Agricultural and Food Economics, 9(1),3
Deaconu, A., Berti, P.R., Cole, D.C., Mercille, G., Batal, M., Market foods, own production, and the social economy: How food acquisition sources influence nutrient intake among ecuadorian farmers and the role of agroecology in supporting healthy diets, 2021, Sustainability (Switzerland), 13(8),4410
Saxe-Custack, A., Lachance, J., Hanna-Attisha, M., Dawson, C., Flint Kids Cook: Positive influence of a farmers' market cooking and nutrition programme on health-related quality of life of US children in a low-income, urban community, 2021, Public Health Nutrition, 24(6), pp. 1492-1500
Schoolman, E.D., Do direct market farms use fewer agricultural chemicals? Evidence from the US census of agriculture, 2019, Renewable Agriculture and Food Systems, 34(5), pp. 415-429
Dias, C., Gouveia Rodrigues, R., Ferreira, J.J., Small agricultural businesses' performance—What is the role of dynamic capabilities, entrepreneurial orientation, and environmental sustainability commitment?, 2021, Business Strategy and the Environment, 30(4), pp. 1898-1912
Panzone, L., Di Vita, G., Borla, S., D’Amico, M., When Consumers and Products Come From the Same Place: Preferences and WTP for Geographical Indication Differ Across Regional Identity Groups, Journal of International Food and Agribusiness Marketing, 2016, 28(3), pp. 286–313
Author Response
Response:
Thank you for pointing out these additional references. We have added them all and expanded our discussion section.
Changes made:
- We have added the “Evidence from Washington, D.C.” in the title.
- We have clarified the research questions (p3 line 15-16)
- We have linked the discussion and conclusion with literature (see highlighted parts in the Discussion Section).
- We have read all the suggested reference pieces and added the references as we see fit:
Cicia, G., Furno, M., Del Giudice, T., Do consumers’ values and attitudes affect food retailer choice? Evidence from a national survey on farmers’ market in Germany, 2021, Agricultural and Food Economics, 9(1),3 (cited)
Deaconu, A., Berti, P.R., Cole, D.C., Mercille, G., Batal, M., Market foods, own production, and the social economy: How food acquisition sources influence nutrient intake among ecuadorian farmers and the role of agroecology in supporting healthy diets, 2021, Sustainability (Switzerland), 13(8),4410 (cited)
Saxe-Custack, A., Lachance, J., Hanna-Attisha, M., Dawson, C., Flint Kids Cook: Positive influence of a farmers' market cooking and nutrition programme on health-related quality of life of US children in a low-income, urban community, 2021, Public Health Nutrition, 24(6), pp. 1492-1500 (cited)
Schoolman, E.D., Do direct market farms use fewer agricultural chemicals? Evidence from the US census of agriculture, 2019, Renewable Agriculture and Food Systems, 34(5), pp. 415-429 (Not cited)
Dias, C., Gouveia Rodrigues, R., Ferreira, J.J., Small agricultural businesses' performance—What is the role of dynamic capabilities, entrepreneurial orientation, and environmental sustainability commitment?, 2021, Business Strategy and the Environment, 30(4), pp. 1898-1912 (Not cited)
Panzone, L., Di Vita, G., Borla, S., D’Amico, M., When Consumers and Products Come From the Same Place: Preferences and WTP for Geographical Indication Differ Across Regional Identity Groups, Journal of International Food and Agribusiness Marketing, 2016, 28(3), pp. 286–313 (cited)
Round 2
Reviewer 3 Report
Dear authors, the paper is interesting and has a decent topic. In my opinion it is necessary to make a little effort to improve it.
Entering into the merits: (4.) Discussion and Conclusion, still needs improvement. it is necessary to connect it with the reference literature, highlight its limits and possible future insights
Author Response
Dear Reviewer,
We did another round of revision of the Discussion and Conclusion part, aiming for better integrating literature into the discussions and highlighting the limits of the study. We also added some future research directions. All changes are tracked.
Thank you for your suggestions,
Xiaochu and Lorraine